# Multifunctional Properties of a *Bacillus thuringiensis* Strain (BST-122): Beyond the Parasporal Crystal

**DOI:** 10.3390/toxins14110768

**Published:** 2022-11-07

**Authors:** Argine Unzue, Carlos J. Caballero, Maite Villanueva, Ana Beatriz Fernández, Primitivo Caballero

**Affiliations:** 1Institute of Multidisciplinary Research in Applied Biology-IMAB, Universidad Pública de Navarra, 31192 Mutilva, Spain; 2Departamento de Investigación y Desarrollo, Bioinsectis SL, Plaza Cein 5, Nave A14, 31110 Noáin, Spain

**Keywords:** *Bacillus thuringiensis*, bioassays, toxicity, multifunctional, Coleoptera, plant-parasitic nematode, mite, phytopathogenic fungi

## Abstract

Chemical products still represent the most common form of controlling crop pests and diseases. However, their extensive use has led to the selection of resistances. This makes the finding of new solutions paramount to countering the economic losses that pests and diseases represent in modern agriculture. *Bacillus thuringiensis* (Bt) is one of the most reliable alternatives to chemical-based solutions. In this study, we aimed to further expand the global applicability of Bt strains beyond their spores and crystals. To this end, we selected a new Bt strain (BST-122) with relevant toxicity factors and tested its activity against species belonging to different phyla. The spore and crystal mixture showed toxicity to coleopterans. Additionally, a novel Cry5-like protein proved active against the two-spotted spider mite. In vivo and plant assays revealed significant control of the parasitic nematode, *Meloidogyne incognita*. Surprisingly, our data indicated that the nematocidal determinants may be secreted. When evaluated against phytopathogenic fungi, the strain seemed to decelerate their growth. Overall, our research has highlighted the potential of Bt strains, expanding their use beyond the confinements of spores and crystals. However, further studies are required to pinpoint the factors responsible for the wide host range properties of the BST-122 strain.

## 1. Introduction

*Bacillus thuringiensis* (Bt) is a Gram-positive, ubiquitous, spore-forming bacterium that produces a parasporal inclusion body (crystal) during its stationary phase of growth. The crystal is primarily composed of insecticidal proteins, which have been effectively used to control lepidopteran and coleopteran crop pests in agriculture, as well as mosquito disease vectors [1,2,3]. Bt-based solutions currently represent 1% of the agrochemical market, including insecticides, fungicides, and herbicides [4]. Within the microbial pesticide segment, Bt solutions lead the bacterial product market, with close to 70% of the total share [5].

Bt crystals are mainly composed of Cry and Cyt proteins which are toxic to species of insects of different orders such as Lepidoptera, Diptera, Coleoptera, Hymenoptera, etc. Additionally, Bt synthesizes other water-soluble components and insecticidal toxins during the growth phase, named Vip (vegetative insecticidal protein) and Sip (secreted insecticidal protein), among others [1,3,6]. In the industry, there are some examples of Bt-based products that are manufactured by concentrating the fermentation broth (such as heat spray drying), hence they retain water-soluble components [7]. However, many of the most relevant products today do not incorporate these pesticidal factors since the method of production involves retrieving spores and crystals through centrifugation, leaving behind non-crystal toxins and other factors.

Today, Bt represents a popular biological resource when controlling populations of lepidopteran, dipteran, and coleopteran pests. One of the reasons for this is the specificity of its host spectrum, making it an alternative to the wide spectrum of chemical synthesized pesticides [1]. For instance, the Cry1, Cry2, and Cry9 groups are mainly toxic to lepidopterans [8,9,10]. Analogously, Cry3, Cry7, Cry8, and Cry1Ia (a subgroup of Cry1) have increased activity against coleopterans [11,12,13,14]. For dipterans, Cry4, Cry10, Cry11, and Cyt have been proven as the most relevant so far [15,16,17,18,19,20,21]. Some exceptions to these are the Cry1B, Cry2A, App6, Xpp22, and Mpp51 proteins, among others, with proven activity across various insect orders [8,22,23,24]. However, its potential for becoming a realistic solution against other challenging pest organisms, such as mites, parasitic nematodes, or phytopathogenic fungi, is still being addressed and studies to demonstrate this are still being conducted [25,26,27,28,29,30]. In the case of phytophagous mites, a number of species from the Tetranychidae (Arachnida) family constitute relevant agricultural pests. The most widely distributed species of these types of spider mites is the highly polyphagous and ubiquitous *Tetranychus urticae* (two-spotted spider mite) [31]. It is relevant in the most extensive crops, but their damage is more severe in vegetable and ornamental crops, which are cultivated in greenhouse conditions [32]. Although detailed information on the damage caused by PPNs (plant parasitic nematodes) is not easily accessible, the current estimations place the world’s economic losses due to these types of pathogens at USD 75–125 billion per year [33]. One of the most relevant PPNs are the root-knot nematodes (RKN) (*Meloidogyne* spp.), which are widespread geographically and have a large host range. For instance, *M. incognita* can parasite more than 2000 species of plants [34,35]. Another significant source of agriculture losses in crops, such as wheat, rice, maize, potato, and soybean, are phytopathogenic fungi [36]. Phytopathogenic fungi can damage the plant (aerial or root part) or affect post-harvesting products. The main fungi species that cause disease in agriculture are *Alternaria* spp., *Fusarium* spp., *Rhizoctonia solani*, *Puccinia* spp., *Sclerotinia* spp., *Phytophthora infestans*, and *Verticillium dahliae*, among others [36,37]. The particularity of some of these organisms is that their control relies on the use of synthetic chemical products [34,38].

Here, we focused on finding and characterizing a wild-type Bt strain as a source of toxicity factors to confer protection and control against diverse pests and pathogens that produce significant economic losses in agriculture. Specifically, its microbial pesticide properties against the Colorado potato beetle (*Leptinotarsa decemlineata*), the two-spotted spider mite (*Tetranychus urticae*), the root-knot nematode *Meloidogyne incognita,* and the *Verticillium dahliae* and *Fusarium oxysporum* species of phytopathogenic fungi were evaluated. Our study showed that the spores and crystals contain toxicity factors with significant activity against coleopterans and mites in vivo. However, the selected strain showed nematocidal properties in vivo and in cucumber plants in the pot assay, which were found to reside within the supernatant after the fermentation process. Additionally, it decelerated the growth of diverse phytopathogenic fungal species in plate assays. Although these preliminary results, under controlled conditions, could represent a good starting point in understanding the potential of strain BST-122, further studies would be required to address the precise factors, mechanisms of action, and overall biosafety of the microorganism. All in all, the results suggest that the potential of Bt strains for controlling different pests may be overlooked by the currently available commercial solutions, which are usually based on mixtures of spores and crystals. Uncovering additional secreted factors and understanding their mechanism of action may prove a reliable source for expanding their uses in the crop protection sector.

## 2. Results

### 2.1. Selection of a Bt Strain Harboring Previously Described Toxicity Factors against Pest Organisms Belonging to Different Phyla

The currently available Bt-based solutions often focus on an insect order at once. Meaning that a Bt solution aimed at controlling dipterans may not be effective for the control of lepidopterans and vice versa. One of the objectives of this study was to find a Bt strain with multifunctional properties in order to control species of phytophagous or plant parasitic pests belonging to different orders, phyla, or even kingdoms of life. In order to search for the right candidate, we screened our library of sequenced Bt strains using the crystal genes *cry5*, *app6*, *cry12*, *cry13*, *cry14*, *cry21*, and *xpp55*, and secreted factors—such as chitinases and metalloproteases—as the main criterium. The reasoning behind this selection was as follows: Cry5, App6, and Cry12, had previously been described as nematocidal/acaricidal proteins [25,26,27,28,39,40], Cry14 and Xpp55 as active against nematocidal and coleopteran [14,25,39], and Cry13 and Cry21 as nematocidal proteins [39,40,41]. Additionally, chitinases and metalloproteases had been reported by other research groups to enhance their activity against nematodes and to possibly affect fungi [42,43,44].

Out of the screened Bt strains from our laboratory library, BST-122 (isolated from a dust sample) was selected for this study since it presented an interesting gene content, with the potential for controlling pests out of the scope of most currently available Bt-based products (Table 1). Specifically, we found the sequence of a cry5-like gene (96% identical to Cry5Ad1 (ABQ82087.1) using local alignment (BLASTP), but with global alignment the identity dropped sharply to 36% pairwise identity and endotoxin_N and delta_endotoxin_C domain-containing protein), which could potentially represent a novel protein for the control of mites and nematodes. Moreover, sequences that codified for the coleopteran- and hemipteran-specific toxins—Mpp51 (ADK94873.1) and ORF2 of cry65Aa (Orf2_cry65Aa; AEB52308.1)—and secreted factors, such as an exochitinase (AIE34993.1) and metalloproteases (Bmp1 (AFZ77001.1) and ColB (ACZ37253.1)), were also present (Table 1). To further characterize the strain, we grew it in CCY medium at 28 °C and 200 rpm, for 72 h. The isolate produced parasporal crystals consisting of two bodies that were attached to the spores. One appeared as a dark, round shape and the other as a dark, bar-shaped crystal (Figure 1A). To further analyze which of the biocidal genes (Table 1) were expressed in the tested conditions and, as a result, integrated into the crystals, the mixture of spores and crystals was run in an SDS-PAGE. As Figure 1B shows, the main three characteristic bands observed in the protein profile of strain BST-122 were approximately 120, 58, and 34 kDa, which correlated with the predicted size of the individual Cry5-like, Orf2_cry65A, and Mpp51Aa crystal proteins, respectively.

In addition to the morphology and crystal protein content and their expression, we tested the BST-122 Bt strain for the production of type I β-exotoxin, which is nonspecific and toxic to vertebrates. HPLC analyses confirmed the lack of this determinant in the BST-122 strain (Appendix A).

To understand the potential of strain BST-122 as a broad microbial control agent we decided to evaluate the activity of its spores and crystals against a model species for each of the potential target orders of organisms.

### 2.2. The BST-122 Bt Strain as a Biological Control Agent against Newly Hatched Larva of Leptinotarsa decemlineata

After analyzing the BST-122 crystals in SDS-PAGE (Figure 1B), we found bands corresponding to the predicted Mpp51Aa protein, indicating that it would be actively expressed by the Bt strain. Mpp51Aa was previously reported as active against coleopterans, among other insect orders [25,45]; therefore, we decided to test the biological activity of the BST-122 wild-type strain against a well-known representative of this order, namely the Colorado potato beetle (*L. decemlineata*). For this purpose, we carried out bioassays in which we addressed the mortality of first-instar larvae, which had fed from superficially contaminated potato leaves, at different concentrations of the BST-122 spore and crystal mixture (s + c). Mortality was registered 4 days post-treatment and the mean lethal concentration (LC_50_) value was calculated from three independent biological replicates. Table 2 shows the effectiveness of the treatments, with a calculated LC_50_ of 10.5 μg/mL.

### 2.3. Activity of BST-122 Bt Strain against Tetranychus urticae Protonymphs

The other major toxicity factor that was found within the BST-122 crystals was a Cry5-like protein. Sharing only 36% of identity with protein Cry5Ad1, this would constitute an entirely new protein, with the potential to control mites and nematodes [26,28,39]. To test the biological relevance of the BST-122 crystals against both types of potential hosts, we first conducted in vivo assays against *T. urticae*. A single concentration of a spores and crystals mixture (100 μg/mL) was fed to protonymphs of *T. urticae* on a Petri dish. The treatment was mixed with a saccharose and blue food dye (fluorella blue) and placed between two layers of Parafilm^®^, which covered the plate. In this manner, mites ingested the treatment by piercing through the parafilm with their sucking/feeding system. After 16 h of contact with the treatment, blue-colored mites were selected and placed on a bean leaf disc. Mortality was recorded after 3 days of treatment ingestion. However, the BST-122 did not present significant mortality compared to the control (data not shown).

Since the mixture of spores and crystals was ineffective against *T. urticae*, we speculated that purifying the Cry5-like protein would help increase its relative concentration and address its true potential in controlling mites. The *cry5*-like gene is found in an operon architecture, alongside the *orf2cry65Aa* gene in BST-122. Therefore, we constructed plasmid pSTAB-Cry5- orf65, which expressed the proteins Cry5-like and Orf2_Cry65Aa under the *cyt1A* promoter (P*cyt1A*) and used it to transform the BMB171 acrystalliferous strain. The resulting strain, BMB171 pSTAB-Cry5-orf65, was grown until the stationary phase, and the mixture of spores and crystals was retrieved by centrifugation. In order to verify the correct expression of both proteins, we ran an SDS-PAGE of the newly generated crystals and found that their predicted molecular masses—119.5 kDa and 58.4 kDa—correlated with the observed bands (Figure 2).

We then carried out in vivo bioassays following the same methodology to test the efficacy of the purified Cry5-like and Orf2_Cry65 crystals. In this case, applying a high concentration—as in the previous experiment (100 μg/mL)—resulted in 53.3% mortality against the two-spotted spider mite nymphs, 72 h after the treatment (Table 3). Strain BMB171, carrying an empty pSTAB vector, was used as a negative control.

### 2.4. The BST-122 Bt Strain as a Biological Control Agent against Meloidogyne incognita J2 Juveniles

To evaluate the potential nematocidal activity of the BST-122 strain on plant parasitic nematodes, we chose *M. incognita*, the most economically important PPN species in the tropical and subtropical regions. For this purpose, we performed bioassays on J2 individuals and evaluated its activity seven days post-treatment. We tested solubilized proteins of the s + c mixture against J2, at different concentrations. The mortality of juveniles registered a tendency dependent on the protein concentration (Appendix A and Appendix A). Since the solubilized protein of the wild-type strain performed below our expectations against *M. incognita*, we decided to test the activity of the Cry5-like protein, following the same reasoning that we applied for *T. urticae*. For this purpose, the BMB171-Cry5-orf65 was tested at two concentrations (50 and 150 μg/mL). Strain BMB171, carrying the pSTAB-empty vector, was used as a control. The results showed relatively high mortality rates for both, the treatment and the control, suggesting that there might be factors that are not present in the crystals that are responsible for exerting toxicity on the nematodes (Appendix A). When comparing the secreted factors of both strains, we found chitinases and metalloproteases Bmp1 and ColB (Appendix A)—with the previously described nematocidal activities—to be present in both strains [42,43,44]. To address the potential toxicity of these factors, we conducted tests in which the supernatant (SN) and the whole culture (WC) were used as treatments. The results showed 61.1% and 52.5% mortality rates, respectively (ANOVA F_3, 8_ = 5.685; *p* = 0.0221 and post-hoc Tukey at *p*-value < 0.05) (Table 4 and Figure 3B). These findings indicated that toxicity factors within the SN may provide BST-122 with the potential to control *M. incognita* populations. Analogously, we tested the effect of the same culture fractions in nematode eggs but found no activity (data not shown).

Our in vivo results suggested that an effective control of *M. incognita* populations may be achieved by using the SN of BST-122 cultures. Since *M. incognita* is an obligate parasite, we decided to corroborate such an observation by performing assays to test the effect of an infestation on cucumber plants. Here, we evaluated the nematocidal activity of s + c, SN, and the WC of our Bt strain when the cultures reached 10^8^ spores/mL (50 μg/mL). As a negative control, water was used. All treatment plants were used 14 days after seeding and infested with 1000 freshly retrieved eggs of *M. incognita*. The experiment was evaluated 28 days after the Bt treatments. A total of three treatments were applied, with a week-long gap between them. The first treatment was implemented a day before the infestation (Figure 4A).

The nematode infection was evaluated by counting the total number of galls in the root parts per plant. The results of the analysis were statistically analyzed using the ANOVA (F_3,78_ = 26.28; *p* = 7.54 × 10^−12^) and the Tukey post-hoc (*p*-value < 0.05) tests. As shown in Table 5, the plants treated with the s + c mixture did not differ significantly, compared to the control (209.2 and 257.6 galls per plant, respectively). However, plants treated with the fermentation supernatant (SN) and with the whole fermentation culture (WC) had a significant drop in the number of recorded galls (112.4 and 76 galls per plant, respectively). Overall, the gall reduction was up to 56.4% in the plants treated with SN and 70.5% in the plants treated with the WC, when compared to the control. This indicated that Bt- secreted factors may play an important role in achieving an effective control of PPNs in plant experiments.

### 2.5. Effect of BST-122 in the Growth of Phytopathogenic Fungi: Verticillium dahliae and Fusarium oxysporum

The above-mentioned results revealed that secreted factors of strain BST-122 may prove useful for the growth inhibition of parasitic nematodes. Phytopathogenic fungi represent another source of organisms that could potentially be controlled by said strain. To address this, we conducted experiments that consisted of comparing the biomass surface of the root phytopathogenic fungi species, *Verticillium dahliae,* and variants of *Fusarium oxysporum* that were grown in LB plates, in the presence or absence of BST-122. Pictures of the growing plates were taken on different days, depending on the fungal species. The results obtained from these experiments are shown in Table 6 and Figure 4. The growth of *Verticillium dahliae* was negatively affected by the presence of the Bt strain, BST-122 (Figure 5A). After 6 days of incubation, a significant slowdown in growth of 24.76% was observed (*t*-test: t = 3.464, df = 19, *p* < 0.01). At 12 days of incubation, this effect was further increased, scoring 63.90% *(t*-test: t = 7.3254, df = 19, *p* < 0.001). Additionally, two different varieties of *Fusarium oxysporum*—*F. oxysporum lycopersici*, which affects tomato plants, and *F. oxysporum melonis*, which infects melon plants—were tested. In the former, the deceleration in growth was significant at 4, 5, 6, and 7 days after the inoculation of the plates, reaching a total biomass surface reduction of 33.72% (*t*-test: t = 6.0417, df = 16, *p* < 0.001) (Figure 5B). In the latter, comparable reduction levels were achieved (30.33%) (*t*-test: t = 4.9758, df = 16, *p* < 0.001) (Figure 5C).

## 3. Discussion

In this study, we selected the strain BST-122, based on its gene content and associated potential as an insecticidal, nematocidal, fungicidal, and acaricidal agent. Some of the most interesting toxins harbored by this strain were a new *mpp51Aa* and a *cry5*-like gene. In a previous study, an Mpp51Aa protein was depicted as active against different insect orders, such as the hemipteran species (*Lygus hesperus* and *L. lineolaris*) and *L. decemlineata*—a coleopteran pest of economic relevance in agriculture [22,46]. In the case of Cry5 proteins, US patents, 5,211,946 and 5,350,576, had attributed Cry5Aa1 and Cry5Ab1 a potential activity against mites belonging to the *Tetranychus* and *Dermatophagoides* genus [26,28] and nematodes (*Caenorhabditis elegans* and the phytoparasitic nematodes, *Pratylenchus* spp. and *M. incognita*) [39,45,47]. Other proteins—such as secreted chitinases and metalloproteinases, namely *colB* and *bmp1* (a type of collagenase)—were also of interest due to their associated nematocidal and fungicidal properties [29,42,43,44,48,49]. Altogether, these characteristics made the selected strain a strong candidate for evaluating its potential multitarget properties.

The next step was testing its overall toxicity against representatives of the different orders and phyla of potential target organisms. One of BST-122’s most relevant crystal proteins was a new Mpp51Aa protein, which could be active against coleopterans [46]. Insects belonging to this order have been traditionally controlled with Cry3 protein variants in formulated Bt-based products, or through their expression in transgenic plants [14]. The lack of diversity in Bt-based biopesticides, as well as the extensive use of engineered plants, have contributed to the emergence of resistant populations [50,51]. For this reason, promoting the use of Bt strains with novel or less commonly used insecticidal toxins is an interesting strategy towards diminishing this problem. Here, we propose strain BST-122 as an alternative to Bt strains harboring Cry3 for the control of *L. decemlineata*. Although we did not determine the activity of the new Mpp51Aa protein, Mpp51Aa1 is known for being active against the Colorado potato beetle (LC_50_ = 19.5 μg/mL) [22,46]. Based on this previous information, we evaluated the activity of a mixture of spores and crystals from BST-122 (LC_50_ of 10.5 μg/mL) (Table 2). Since Mpp51Aa seems to be present in the BST-122 crystals, one would assume that this protein could contribute to its overall activity against *L. decemlineata* (Figure 2). However, other proteins present in the crystal, such as Cry5-like, Orf2_cry65A, or Mpp2-like, could represent an additional source of toxicity or interact synergistically with the Mpp51Aa protein.

Another crystal protein of interest that is expressed by BST-122 was a Cry5-like candidate. Although there is little information available on the activity of this protein against mites, two patent publications on its potential acaricidal activity made us consider whether it might be possible to use BST-122 to control *T. urticae* populations [26,28]. Phytophagous mites have a particular feeding system that differs from the coleopteran or lepidopteran insect larvae. They typically feed on leaf tissue by inserting a stylet into it and sucking the epidermic and parenchymatic cell content. As a result, the plant cells collapse and die, causing chlorotic spots on the leaves and reducing the rate of transpiration and photosynthesis in the plant [32,52]. In this study, the characterization of Bt BST-122 as a potential acaricidal source was evaluated under in vivo conditions. For this purpose, we developed a new assay method that consisted of placing *T. urticae* nymphs in a Petri dish and providing spore and crystal mixtures of the selected strain in a drop containing a sucrose solution and food dye. After 16 h, colored nymphs were selected and placed on a bean leaf. The mortality of individuals was registered 72 h after treatment ingestion. Although the BST-122 strain did not reveal toxicity in mites, when testing the Cry5-like protein alone, at the same rate, 53.3% of mortality was recorded (Table 3). Most available studies on Bt acting as a control agent against mites have mainly focused on the activity of wild-type strains. For example, the Bt strains EA3, EA11.3, and EA26.1 were reported as being considerably active (LC_50_ of 7.111, 12.839, and 1.509 μg/mL, respectively) against the honeybee ectoparasite, *Varroa destructor* [53]. Other studies reported that strain GP532 was effective against the rabbit ear mite ectoparasite, *Psoroptes cuniculi* [54,55]. Additionally, strain BPU5 has been described as displaying acaricidal properties against the phytophagous mite, *Tetranychus marcfarlanei,* at relatively high concentrations (LC_50_ = 8.024 mg/mL) [56]. However, few publications have described the factor responsible for the acaricidal properties of Bt. Some research studies have described β-exotoxin as acaricidal against the following phytophagous mites: *Panonynchus citri*, *P. ulmi*, *Tetranychus telarius*, *T. urticae*, *T. pacificus,* and the predator, *Metaseiulus occidentalis* [57,58,59,60,61]. Nonetheless, the β-exotoxin is toxic to mammals and, therefore, strains expressing such a determinant are not eligible for product development. Regarding delta-endotoxins, only US patents 5,211,946; 5,350,576; and 5,262,158 mention Cry5Aa1, Cry5Ab1, App6Aa1, App6Ba1, Cry8Aa1, and Cry12Aa1 as potentially active against the two-spotted spider mite (*T. urticae*) and the house dust mite, *Dermatophagoides pteronyssinus* [26,27,28]. Regardless, experiments have been conducted using the wild-type strains, expressing crystals with the said combinations of toxins, and significant results have only been obtained when applying high concentrations of the active ingredients (5 mg/mL). As opposed to the publicly available data, our results showed significant mortality when we used a novel protein against *T. urticae* nymphs, at a relatively low dose (100 µg/mL). However, the tested concentration was still too high from a cost-effective standpoint. Possibly, the most interesting contribution of these results is the finding of a new protein with acaricidal properties that could serve as a basis for searching new Cry5-like proteins with improved activity. Additionally, a foliar application would not be effective for the control of phytophagous mites due to their feeding behavior. Therefore, opting for the expression of Cry5-like candidates in the vascular system of plants may represent a more realistic approach for the control of these pests in agriculture.

Furthermore, Cry5 proteins had previously been described as potentially nematocidal [25,39]. This made us consider whether it would be effective against nematodes of agronomic importance. To test the potential of the strain, we performed in vivo assays using second-stage juveniles of the phytoparasitic nematode, *M. incognita*. The physiology of plant parasitic nematodes should be considered, whose feeding systems consist of a stylet, with a diameter of only 28 kDa for the cyst nematode, *Heterodera schachtii* [62]. However, it is known that solubilized proteins of up to 140 kDa can be incorporated into the gut of *Meloidogyne hapla* [63]. For this reason, we solubilized BST-122 crystals to test their activity against *M. incognita* J2 juveniles. After exposing individuals to increasing concentrations of BST-122 solubilized crystals, we found a correlation between the concentration and mortality (Appendix A). However, significant mortality was only reached at a relatively high concentration. Following the same strategy as with *T. urticae*, we decided to test whether the purified Cry5-like protein had increased activity against J2 individuals. The results showed that treatments with the Cry5-orf65 crystal expressed from the BMB171 strain were undifferentiated from the negative control, the BMB171 strain carried an empty plasmid (Appendix A). This suggested that the acrystalliferous BMB171 strain may harbor virulence factors that are not related to crystal pesticidal proteins (Appendix A). This was in agreement with previous reports, in which BMB171 was attributed nematocidal properties [44,64]. Additionally, other studies addressed how the activity of purified crystals containing App6Aa and Cry5Ba in *C. elegans* was improved (4.73-fold and 3.59-fold, respectively) in the presence of the spores. It was suggested that the ColB metalloproteinase could be involved in nematocidal pathogenesis [44]. In a similar manner, a previous study demonstrated the synergistic activity of the Bmp1 metalloproteinase when mixed with Cry5Ba in *C. elegans* [43]. These studies indicated that Bt may produce factors that are not present in the crystal and that may represent an alternative to δ-endotoxins in the field of microbial pesticides. Therefore, since BST-122 shared many of the BMB171-secreted factors (Appendix A), we considered whether these would be enriched in the supernatant and, perhaps, exert a greater nematocidal activity than the proteins of the crystal alone. To address this idea, we conducted treatments with the supernatant and the whole fermentation culture of the BST-122 strain against *M. incognita* juveniles. The results showed increased activity for both treatments when compared to the spore and crystal mixture. This indicated that the supernatant contained one or more toxicity factors that were responsible for the observed mortality (Table 4 and Figure 3).

Since *M. incognita* is an obligate parasite, we performed experiments treating the radicular system of nematode-infested cucumber plants to validate the in vivo results. When applying similar concentrations of the aforementioned treatments to cucumber plant pots, a reduction of 56.4% and 70.5% of galls per plant was observed for the fermentation supernatant and the whole culture, respectively (Table 5 and Figure 4). This was consistent with our in vivo observations and with studies from other research groups, in which the activity of Bt strains, ToIr65 and ToIr67, against *M. javanica* in tomato plants were more effective when the fermentation supernatant was applied [65]. Moreover, other studies, in which the whole culture, the fermentation supernatant, and the spores and crystals of diverse Bt strains were tested against *M. incognita,* showed how the application of the whole culture reduced the number of eggs by 84% [66]. These studies did not specify the main factors behind the observed toxicity. However, other research studies did indicate that the nematocidal properties of Bt supernatants may be due to the presence of β-exotoxin [67,68,69]. Although the toxicity of β-exotoxin in mammals is yet unclear, the public use of this active substance is forbidden by the recommendation of the World Health Organization [70]. In this study, the possibility of I β-exotoxin (thuringiensin) being responsible for the observed mortality and gall reduction in plant assays was excluded since its presence could not be detected by genome sequencing nor by HPLC analysis.

Some of the BST-122 secreted factors involved chitinases. These had been previously described as potential antifungal agents [29,49]. This made us consider whether BST-122 may also prove useful in controlling some of the most relevant phytopathogenic fungi in agriculture. To characterize the potential fungicidal activity of BST-122, we evaluated the growth inhibition of several fungal species that cause plant disease in the presence of the BST-122 strain. The experiment consisted of inoculating LB agar plates with an agar disc of each fungus flanked by two drops of a BST-122 culture (10^6^ CFU/mL) to evaluate the effects. The tested species were *Verticillium dahliae* and two different serovars of *Fusarium oxysporum* (*lycopersici* and *meloni*). The fungal growth was measured at different moments of incubation, depending on the species. The in vivo growth inhibition was 60% in *V. dahliae* and approximately 30% in *F. oxysporum,* when compared to the control (Table 6 and Figure 5). These results suggest that the presence of the bacterium may decelerate the growth of the tested fungi species. Nonetheless, further experiments would be required to address the overall potential fungicidal activity of BST-122—for instance, by infecting plant tissue to evaluate the damage caused by the phytopathogenic fungi when treated with BST-122. Several compounds produced by the *Bacillus* spp. have been described as active against fungal phytopathogens [71]. In a previous study, isolates of *Bacillus haloterans* exhibited strong activity against diverse *Fusarium* sp.—*Botrytis cinerea*, *Alternaria alternata*, *Phytophthora infestans*, and *Rhizoctonia bataticola* [72]. In the particular case of *B. thuringiensis*, different compounds have been demonstrated as being beneficial for the control of phytopathogenic fungi. In another report, volatile organic compounds produced by the BCN10 strain resulted as antifungal in vitro and in vivo against five postharvest pathogens, such as *F. oxysporym* [73]. Other studies have revealed that chitinase extracted from Bt strains could significantly inhibit the mycelial growth of several pathogenic fungi [29,49]. Moreover, the fungicidal activity of diverse Bt strains has been attributed to the production of some peptides, for instance, the fengycin-like lipopeptides [74,75,76].

To summarize, we have characterized the potential of Bt strain BST-122 for the control of diverse plant pests and diseases, such as coleopteran pests, plant-parasitic nematodes, phytophagous mites, and plant pathogenic fungi—which are organisms that cause significant economical losses in agriculture worldwide. However, further studies on BST-122 are required to pinpoint the toxicological factors responsible for controlling each of the tested hosts; evaluate their activity against a broader host spectrum (for instance, hemipteran pests) analyze their impact on beneficial insects—such as pollinators or natural enemies—and verify soil ecosystem and agricultural compatibility. Overall, this work contributes to highlighting the importance of the multitarget potential of Bt strains, a notion that goes beyond the spores and crystals and that takes into consideration additional factors that could extend the use of Bt products to pests and diseases that are currently treated with synthetic chemicals.

## 4. Materials and Methods

### 4.1. Selection of Bacterial Strains

#### 4.1.1. Bacterial Isolation

A laboratory library of Bt strains was built from diverse substrate samples, belonging to different geographical locations in Spain. Samples were mixed with sterile dH_2_O, incubated at 72 °C for 20 min, and serial dilutions plated onto CCY agar plates [77,78]. Next, plates were incubated for 48 h, at 28 °C, and single colonies were inspected under an optical microscope to confirm the presence of parasporal crystals. Positive single colonies were cultured until cell lysis, and stored at −20 °C.

#### 4.1.2. Total DNA Extraction and Genomic Sequencing

Total genomic DNA (chromosomal + plasmid) was extracted from the isolated strains, following the protocol for DNA isolation from Gram-positive bacteria supplied in the Wizard**^®^** Genomic DNA Purification Kit (Promega, Madison, WI, USA). The DNA library was prepared from total DNA and subsequently sequenced by Illumina NextSeq500 Sequencer (Genomics Research Hub Laboratory, School of Biosciences, Cardiff University, Cardiff, UK).

#### 4.1.3. Identification of Potential Nematocidal/Insecticidal, Acaricidal, and Fungicidal Genes

The genomic raw data were processed and assembled using CLC Genomic Workbench v10.1.1 (Aarhus, Denmark). Reads were trimmed and filtered, and those shorter than 50 bp were removed. Processed reads were assembled de novo using a stringent criterion of overlap of at least 95 bp of the read and 95% identity, and reads were then mapped back to the contigs for assembly correction. Genes were predicted using GeneMark v2.5 (Atlanta, GA, USA) [79].

To assist the identification process of potential pesticidal toxin proteins, local BLASTP [74] was executed against a database built in our laboratory, including the amino acid sequences of known bacterial toxins from the bacterial pesticidal protein database (https://camtech-bpp.ifas.ufl.edu, accessed on 18 July 2022) [80,81], as well as other proteins of interest such as chitinase, enhancin-like metalloprotease, Bmp1, CalY, ColB, and InhA proteins. Proteins involved in the synthesis of zwittermycin A and β-exotoxin were also included [82]. Prediction of structurally conserved domains was carried out using CD-search [83]. Pairwise sequence alignment was carried out using GGSEARCH2SEQ v36.3.8h (Cambridgeshire, UK.) [84].

#### 4.1.4. Production of Spores and Crystals from the Wild Bt Strain

For BST-122, single colonies from LB plates were inoculated in 50 mL of CCY sporulation culture medium [78], and grown under constant temperature (28 °C) and shaking (200 rpm). Crystal formation was observed daily under an optical microscope, at the magnification of ×1000 (Zeiss Axiolab 5, Carl Zeiss Microscopy GmbH, Jena, Germany). After 48 or 72 h, when approximately 95% of the cells had lysed, cultures were stored at 4 °C, until required. The number of spores/mL at the end of the fermentation process was addressed by plating serial dilutions in LB agar Petri dishes. The supernatant and the mixture of spores and crystals were collected by centrifugation at 9000× *g*, at 4 °C, for 10 min. The supernatant was stored at 4 °C. The spore and crystal mixture, after being washed with a saline solution (1 M NaCl, 10 mM EDTA), was resuspended in dH_2_O, and kept at 4 °C. For protein quantification, the spore and crystal mixture was solubilized in carbonate buffer (50 mM Na_2_CO_3_, 100 mM NaCl, pH 11.3) and 10 mM DTT, for two hours, at 37 °C, and total protein concentration was quantified by Bradford assay [85], using bovine serum albumin (BSA) as a standard.

#### 4.1.5. SDS-PAGE

Samples of spores and crystals, as well as solubilized proteins, were mixed with 2× sample buffer (Bio-Rad Laboratories Inc., Hercules, CA, USA), boiled at 100 °C for 5 min, and then subjected to electrophoresis, using Criterion TGX TM 4–20% Precast Gel (Bio-Rad Laboratories Inc., Hercules, CA, USA) [86]. Gels were stained with Coomassie Brilliant Blue R-250 (Bio-Rad Laboratories Inc., Hercules, CA, USA) and then distained with 30% ethanol and 10% acetic acid solution.

### 4.2. Detection of β-Exotoxin

The presence of type I β-exotoxin (thuringiensin) was evaluated in culture supernatants through HPLC analysis (Department de Gènetica, Universitat de València, Burjassot, Spain) [87]. The standard strain, HD-2 strain, was used as a positive control.

### 4.3. Coleopteran, Acaricidal, Nematocidal, and Fungicidal Activity of BST-122 Bt Strain

#### 4.3.1. *Leptinotarsa decemlineata*, *Tetranychus urticae*, and *Meloidogyne incognita* Rearing

A laboratory colony of *L. decemlineata* was established from adults collected from organic potato fields near Pamplona (Spain) and maintained on potato plants (*Solanum tuberosum* L. cv. Jaerla). The population of *M. incognita* and *Tetranychus urticae* were obtained from the company Koppert-Spain (Almería, Spain). Nematodes were maintained on tomato plants (*Lycopersicum esculentum,* Mill. cv. Roma). Mites were maintained on bean plants (*Phaseolus vulgaris* L. cv. Garrafal Oro). All populations were kept in a phytotron, under controlled conditions of temperature, humidity, and photoperiod (25 ± 1 °C, 70 ± 5% RH, and L16:D8 h). Nematode eggs were extracted from the root-knot of at least 6-week-old infected roots [88]. The eggs were collected and rinsed with distilled water in sieves (25 and 20 μm pore). To collect the second-stage juveniles (J2), the eggs were placed in hatching dishes and incubated in moist chambers, in distilled water at 25 °C, in darkness. Freshly hatched J2 were collected every 48 h and used for experiments.

#### 4.3.2. Phytopathogen Fungi Species Maintenance

Different phytopathogen fungi species were provided by the Regional Diagnostic Center of the Regional Government of Castilla y León (Salamanca, Spain). The tested fungi were *F. oxysporum* sv. *lycopersici* and *F. oxysporum* sv. *melonis* as root-pathogens, and *V. dahliae*, as an aerial plant pathogen. Fungi were grown on potato dextrose agar (PDA) medium (Sigma-Aldrich, St. Louis, MO, USA), and incubated at 28 °C.

#### 4.3.3. *L. decemlineata* Bioassays In Vivo

The bioassays were performed by the leaf dip method on first instar larvae [89]. Five different spore and crystal mixture concentrations, ranging from 1.8 to 150 μg/mL, were prepared to determine the concentration–mortality responses and calculate the mean lethal concentration (LC_50_). This bioassay was repeated at least three times. Total insect mortality was recorded after 4 days.

#### 4.3.4. *T. urticae* Bioassays In Vivo

The bioassays were performed with first instar nymphs of *T. urticae*. For each treatment, 20–25 nymphs were placed in 35 mm diameter Petri plates, under a dissecting microscope (Zeiss Stemi 508, Carl Zeiss Microscopy GmbH, Jena, Germany) and the plates were covered with Parafilm^®^. A 200 μL drop of treatment (100 μL of Bt spore and crystal mixture, 80 μL of 79% of sucrose w/v, and 20 μL of food dye, diluted in sucrose solution) was placed between two layers of Parafilm^®^. The mites were incubated at 25 °C, under light conditions, during 16 h, and the colored nymphs were placed on a bean leaf disc (20 mm diameter, on a 2% agar plate). The mortality was scored 72 h after the ingestion. Bt treatments were tested at a single dose of 100 μg/mL. A mixture of sucrose and food dye was used as control.

#### 4.3.5. *Meloidogyne incognita* Bioassays In Vivo

##### J2 Mortality Treated with Spores and Crystal Mixture, Supernatant, and the Whole Culture

The spore–crystal mixture (s + c), supernatant (SN), and the whole culture of the BST-122 Bt strain were tested against *M. incognita* J2 juveniles. All three different fractions came from a culture grown at 28 °C and 200 rpm, until a concentration of 50 μg/mL (10^8^ spores/mL) was reached. This concentration of Bt strains was prepared by adding the appropriate volumes of 20 mM HEPES (pH 8.0) to the standard solution. A direct contact assay was carried out in a 96-well plate by modification of the standard method [67]. A total of 200 μL of each treatment was added to 5 μL of a J2 nematode suspension, containing a minimum of 20 individuals per well. Treatments were prepared in triplicate and incubated at room temperature. A solution containing 20 mM HEPES (pH 8.0) was used as a control. All treatments were replicated three times. The number of active and dead J2 were counted 7 days after the treatment, under an inverted microscope (Zeiss Primovert, Carl Zeiss Microscopy GmbH, Jena, Germany), and mortality percentages were calculated.

##### J2 Mortality Treated with Solubilized Protein

The spore and crystal mixtures were collected and washed, as described above. The spore and crystal mixture pellet was resuspended in solubilization buffer (0.05 M Na_2_CO_3_, 0.1 M NaCl, pH 11.3 + 0.01 M DL-Dithiothreitol) and incubated at 37 °C, for two hours. Insoluble debris were removed by centrifugation and the solubilized proteins from the supernatant run in SDS-PAGE. This solubilization was dialyzed in 20 mM HEPES (pH 8.0), with a 10 kDa pore Slide-A-Lyzer^®^ Dialysis Cassette (Thermo-Scientific, Rockford, IL, USA) (adapted from [90]). The protein quantification was performed as described above. Different concentrations of the soluble crystal/spore protein of Bt strains, ranging from 50 to 400 μg/mL, were prepared by adding the appropriate volumes of 20 mM HEPES (pH 8.0) to the standard solution. The assay was conducted as described above. A solution containing 20 mM HEPES (pH 8.0) was used as a control treatment.

#### 4.3.6. Assay of Control of *M. incognita* Infestation in Cucumber Plants

Surface-sterilized cucumber seeds (*Cucumis sativus* cv. Marketer) were germinated in 200 cc pots, in autoclaved vermiculite. Plants were fertilized with Osmocote**^®^** at the recommended doses and grown in a phytotron, under controlled conditions of temperature, humidity, and photoperiod (25 ± 1 °C, 70 ± 5% RH, and L16:D8 h). Fourteen days after seeding, when the 2nd real leaf had developed, 30 mL of the test solution was added around the root zone as a first treatment (adapted from [68]). 24 h later, plants were inoculated with a 1 mL suspension, containing approximately 1000 freshly extracted eggs of *M. incognita* (adapted from [91]). Treatments were repeated 7 and 14 days after the first application (Figure 4A). Bt strains were tested at a concentration of 10^8^ spores/mL (adapted from [92,93]). The nematicide potential of each Bt strain was checked, testing the effectiveness from the whole culture (WC), the spore and crystal mixture (s + c), and the supernatant (SN). dH_2_O was used as a negative control. Irrigation with dH_2_O was conducted 3 times per week. Plants were harvested 28 days after the first treatment, and symptoms of nematode infection, namely gall formation on the roots, were recorded 28 days after the first treatment. The experiment was reproduced 4 times, with 5 technical replicates for each biological repetition.

#### 4.3.7. Phytopathogen Fungi Species Bioassays In Vitro

The Bt strain was grown from a single colony, in LB medium, in an overnight culture, at 28 °C and at 200 rpm. Treatments were prepared at a final concentration of 10^6^ CFU/mL, in a 0.85% NaCl solution, and two drops of 5 μL were placed flanking a 7 mm diameter fungal disc on an LB plate. As a negative control, LB plates inoculated with fungal disks were used. All plates were incubated at 28 °C until the negative control grew around the surface of the plate (methodology adapted from [72]). Fungal growth measures were analyzed using the ImageJ (v1.53a) software (https://imagej.nih.gov/ij/download.html, accessed on 15 March 2022). The percentage of growth inhibition was calculated using the following equation: I = (C−T)/C × 100, where I = percent growth inhibition; C = growth in control; T = growth in treatment [94]. All treatments were tested per triplicate.

### 4.4. Recombinant Protein Expression, Purification, and In Vivo Testing

#### 4.4.1. Amplification and Cloning of the cry5-orf2cry_65 Operon Genes

For the construction of plasmids expressing Bt toxins, the *cry5*-*orf2cry65* was first amplified by PCR from the BST-122 Bt strain genomic DNA, using the Phusion High-Fidelity DNA polymerase (NEB, Ipswich, MA, USA) and the corresponding primers harboring restriction enzyme recognition sequences in their extremes (Appendix A). The resulting PCR products were purified by the NucleoSpin**^®^** Gel and PCR Clean Up kit (Macherey-Nagel Inc., Bethlehem, PA, USA), and ligated into the pJET plasmid (CloneJET PCR Cloning Kit, Thermo Scientific, Waltham, MA, USA). The ligation products were then electroporated into *E. coli* XL1 blue cells by using a standard protocol [95]. Colony-PCR was performed to check positive clones from which the plasmids were purified using the NucleoSpin**^®^** Plasmid Kit (Macherey-Nagel Inc., Bethlehem, PA, USA), following the manufacturer’s instructions. Subsequently, the pJET plasmids were verified by sequencing (StabVida, Caparica, Portugal) and digested with the corresponding restriction enzymes to excise the fragment and interest. These were then purified from agarose gels and ligated into a pre-digested pSTABr vector, using the Rapid DNA ligation kit (ThermoScientific, Vilnius, Lithuania), to obtain the recombinant plasmid pSTABr-*cry5-orf2cry65*. The ligation products were then electroporated into *E. coli* XL1 blue cells by using standard protocols [95]. Positive clones were verified by colony-PCR and plasmids were purified and electroporated into the acrystalliferous BMB171 Bt strain.

#### 4.4.2. Production, Purification, and Activity Assay against Diverse Target Species of the Recombinant Toxins

Single colonies from the BMB171-Cry5-orf65 recombinant strain were inoculated in 100 mL of LB culture medium, supplemented with erythromycin, and grown at 28 °C and 200 rpm, for 72 h. The whole was collected by centrifuging at 9000× *g*, at 4 °C, for 10 min, and the pellet was washed with dH_2_O, twice. All protein quantifications were performed by Bradford and tests of the recombinant toxins against protonymphs of *T. urticae* and juveniles of *M. incognita* were carried out as described above. The BMB171-pSTAB-empty strain was used as a negative control in the bioassays.

#### 4.4.3. Nucleotide Sequence Accession Numbers

The nucleotide sequence data reported in this paper have been deposited in the GeneBank database, under the following accession numbers: OP604599 for the *cry5*-like gene; OP696897 for the *mpp51Aa* gene; OP722690 for the *orf2_cry65A* gene; OP722691 for the *mpp2*-like gene; OP722692 for the *exochitinase* gene; OP722693 for the *colB* gene; and OP722694 for the *bmp1*-like gene.

### 4.5. Statistical Analyses

The mean lethal concentration (LC_50_) of the spore and crystal mixture (*L. decemlineata*) and solubilized protein (*M. incognita*) of BST-122 Bt strain were determined based on the probit model [96]. The analysis of BST-122 activity against *M. incognita* J2 mortality (in vivo) and nematode infection in roots was performed by running the ANOVA (*p*-value < 0.05) and Tukey post-hoc tests. For the Cry5-like activity analysis, for *T. urticae* it was performed by Welch’s *t*-test (*p* < 0.05), and for *M. incognita*, as the data were non-parametric, it was performed with the Kruskal–Wallis test (*p* < 0.05), and pairwise comparisons between group levels were performed with Bonferroni correction. The statistical analyses of fungal growth were performed by running Welch’s t-test and the t-test for non-homoscedastic and homoscedastic data, respectively. All analyses were performed with the R (v.4.1.1.) software.

## Figures and Tables

**Figure 1 toxins-14-00768-f001:**
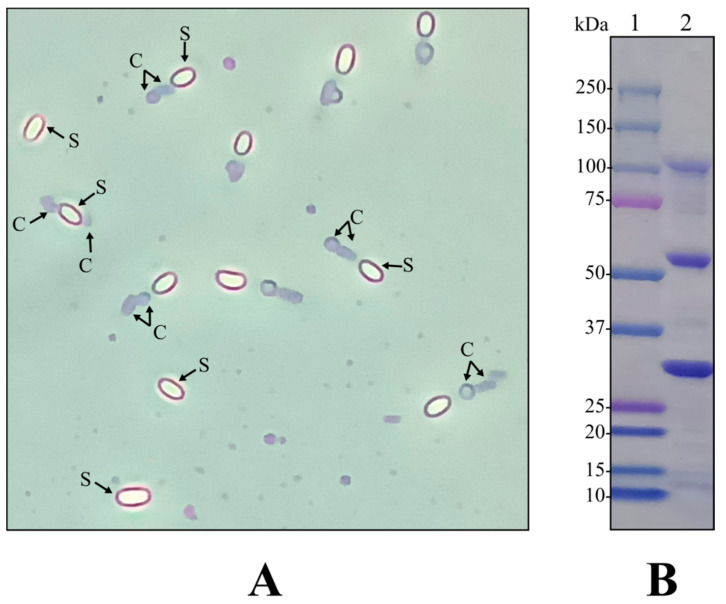
(**A**): picture of the BST-122 Bt strain, as seen under the optical microscope (1000× magnification) after growth in CCY medium for 72 h, at 28 °C and 200 rpm. S—spore; C—crystal. (**B**): SDS-PAGE of BST-122 mixture of spores and crystals. The strain was grown as described in A. In total, 15 μL of the mixture of spores and crystals were denaturalized, mixed with 15 μL of loading buffer for 5 min, at 95 °C. In total, 20 μL of the mix was loaded into the 4–20% polyacrylamide gel and Coomassie staining was applied to reveal discrete bands, corresponding to crystal and spore proteins: line 1—protein marker; line 2—BST-122 wild-type strain.

**Figure 2 toxins-14-00768-f002:**
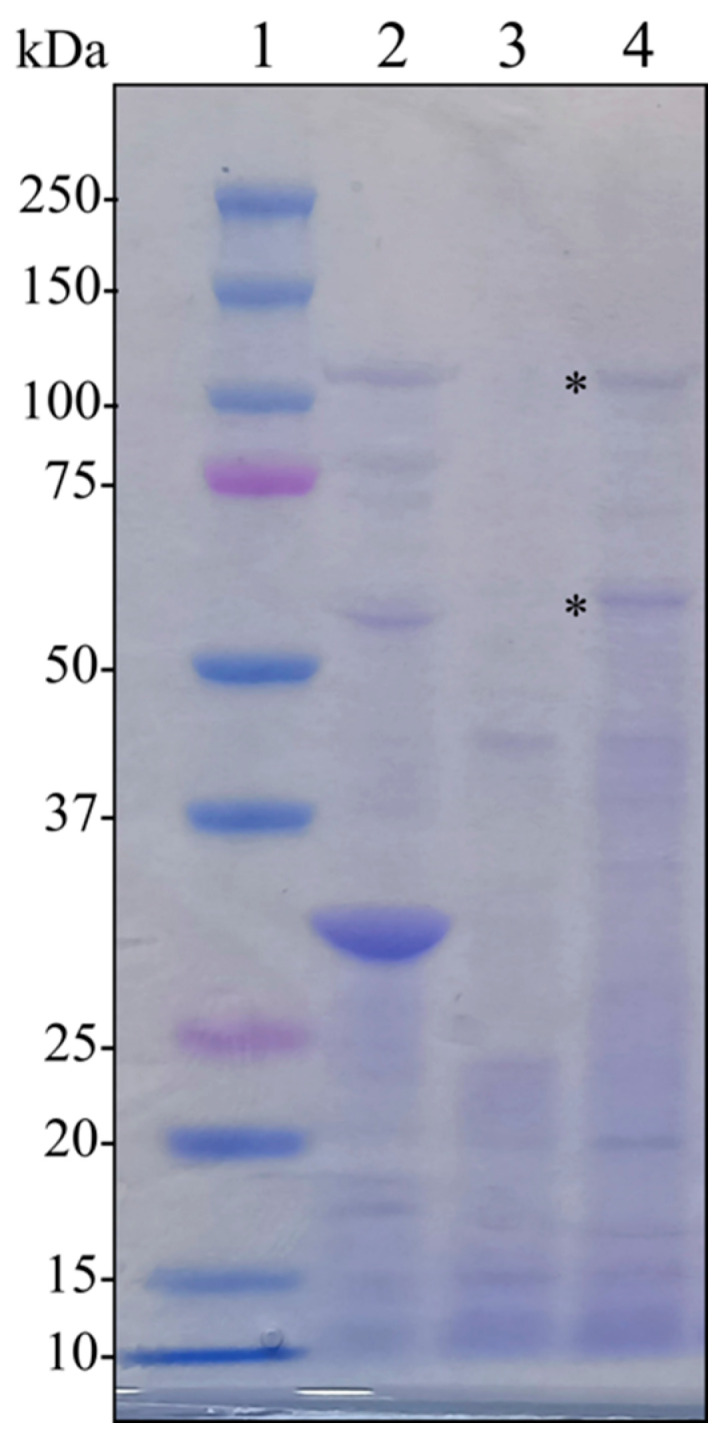
SDS-PAGE of the BST-122 recombinant protein (Cry5-orf65): 1—protein marker; 2—BST-122 wild-type strain; 3—BMB171-empty; 4—BMB171-Cry5-orf65 (* marked bands 119.5 and 58.4 kDa).

**Figure 3 toxins-14-00768-f003:**
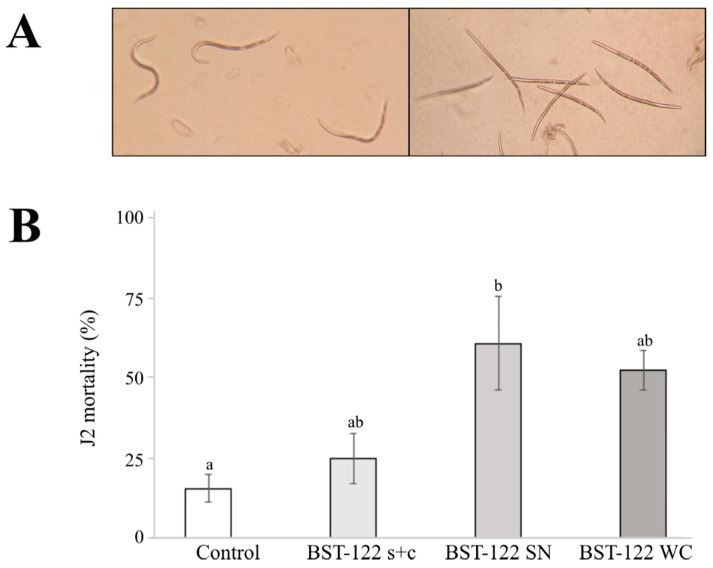
**(A**): pictures of in vivo bioassays of J2 *M. incognita,* as seen under an inverted microscope (100× magnification). Left—alive J2; right—dead J2. (**B**): results of in vivo bioassays of J2 mortality in *M. incognita,* after 7 days treated with the BST-122 Bt strain, at a 50 μg/mL concentration. Control—20 mM HEPES (pH 8.0); BST-122 s + c—mixture of spores and crystals of Bt BST-122 strain; BST-122 SN—supernatant of the fermentation of Bt BST-122 strain; and BST-122 WC—the whole culture of the fermentation of Bt BST-122 strain. Different letters were used to denote statistical significance between values.

**Figure 4 toxins-14-00768-f004:**
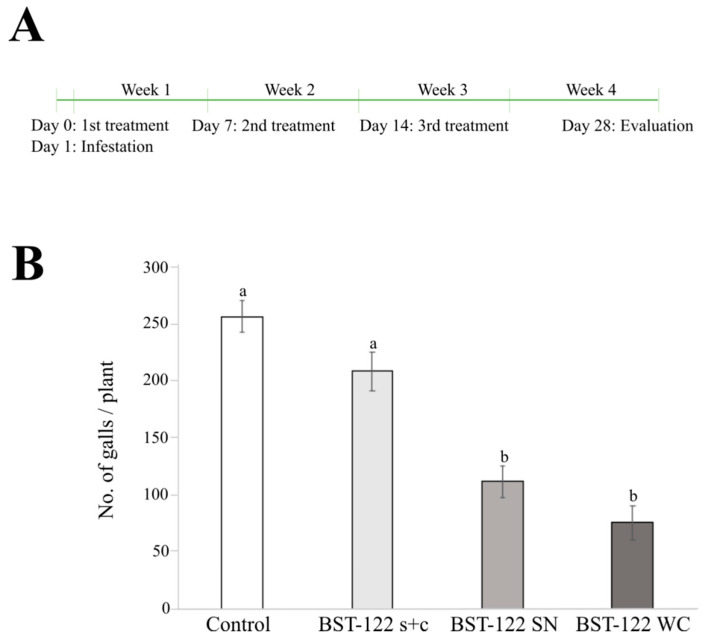
Activity of the BST-122 Bt strain against *M. incognita* eggs, tested in cucumber plants. (**A**): scheme of the methodology established to perform the experiments of BST-122 activity against *M. incognita* in cucumber plants. (**B**): Activity of BST-122 Bt strain against *M. incognita* eggs, tested in cucumber plants. It represents the no. of galls per plant in different treatments. Plants were infested with 1000 freshly laid eggs. Control—infested plants treated with H_2_O; s + c—infested plants treated with the fermentation centrifugation or spores and crystal mixture; SN—infested plants treated with the fermentation supernatant; WC—infested plants treated with the whole fermentation culture. Different letters were used to denote statistical significance between values (ANOVA, followed by post-hoc Tukey test at *p* < 0.05).

**Figure 5 toxins-14-00768-f005:**
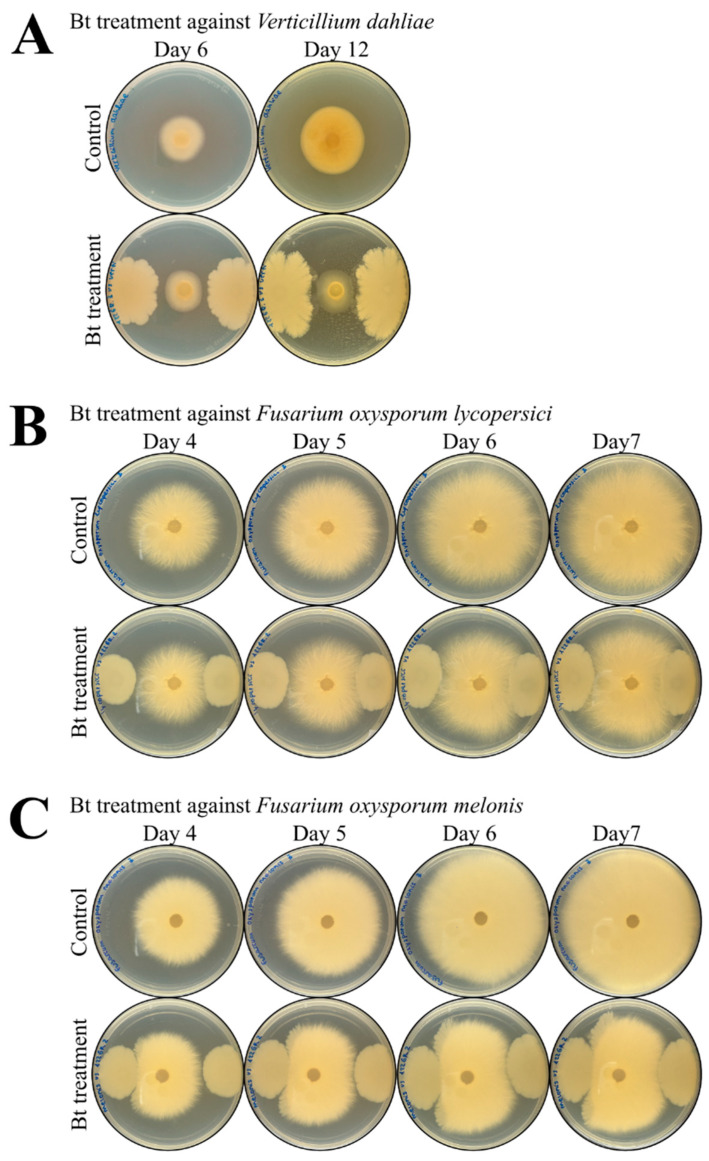
In vitro assays testing BST-122 Bt strain activity against diverse phytopathogenic fungi. The fungi species tested were *Verticillium dahliae* (**A**), *Fusarium oxysporum lycopersici* (**B**), and *Fusarium oxysporum melonis* (**C**). Briefly, a PDA agar disc with the fungus was inoculated in the center of an LB plate, and 5 μL of treatment was inoculated on each side of the fungus. A physiological solution (0.85% NaCl) was used as control, and an overnight culture in LB of the BST-122 strain, adjusted to 10^6^ spores/mL, was used as the tested treatment. Pictures of fungal growth were taken at different development moments and the growth inhibition was measured with ImageJ (v.1.53a) software.

**Table 1 toxins-14-00768-t001:** Insecticidal protein content of the BST-122 Bt strain.

Target Database	Pairwise Identity (%)	MW (kDa)	Length (No. Amino Acid Residues)	Amino Acids Overlap in Global Alignment	Accession Number of Reference	Accession Number
**Crystal proteins**						
Mpp51Aa2	98	34.5	312	312	ADK94873.1	OP696897
Cry5Ad1	36	119.2	1092	1148	ABQ82087.1	OP604599
Orf2_cry65Aa	96	58.4	512	490	AEB52308.1	OP722690
Mpp2Aa6	30	46.2	412	136	U41822.1	OP722691
**Secreted factors**						
Exochitinase	99	39.4	360	360	AIE34993.1	OP722692
ColB (metalloprotease)	77	48.3	426	428	ACZ37253.1	OP722693
Bmp1 (metalloprotease)	34	65.3	592	367	AFZ77001.1	OP722694

**Table 2 toxins-14-00768-t002:** Mean lethal concentration (LC_50_) value of the BST-122 Bt strain, calculated for newly hatched larva of *L. decemlineata*.

LC	Concentration (μg/mL)	Lower Limits	Upper Limits	χ^2^	df	Slope	SE Slope	Intercept
LC_50_	10.5	6.99	15.0	1.96	5	0.947	0.111	−0.969

LC—lethal concentration; χ^2^—chi-square; df—degree of freedom; SE—standard error.

**Table 3 toxins-14-00768-t003:** Acaricidal activity of the BST-122 recombinant protein (Cry5-orf65) at a single concentration (100 μg/mL) on protonymph of *Tetranychus urticae*.

Treatment	Mortality (%)
BMB171-Cry5-orf65	53.3 ± 16.3 a
BMB171-pSTAB	11.7 ± 3.4 b

Percentage of mortality (mean ± standard error) recorded 72 h after treatment. Different letters were used to denote statistical significance between values, Welch’s *t*-test (F_1, 3.224_ = 10.105, *p*-value = 0.04538).

**Table 4 toxins-14-00768-t004:** Mortality of *M. incognita* J2, treated with different fractions of a BST-122 fermented culture.

Treatment	J2 Mortality (%) Mean ± SE
Control	15.5 ± 4.3 a
BST-122 s + c	24.7 ± 7.9 ab
BST-122 SN	61.1 ± 14.7 b
BST-122 WC	52.5 ± 6.3 ab

SE—standard error. Different letters were used to denote statistical significance between values. ANOVA test (F_3,8_ = 5.685; *p* = 0.0221) and post-hoc Tukey test at *p*-value < 0.05.

**Table 5 toxins-14-00768-t005:** Activity of the BST-122 strain against *M. incognita* in cucumber plants. The effectiveness was addressed by evaluating the number of galls per plant after the different treatments.

Treatment	No. of Galls/ Plant ^1^	% Reduction of Galls/Plant
Control	257.6 ± 13.9 a	-
BST-122 s + c	209.2 ± 17.1 a	18.8%
BST-122 SN	112.4 ± 14.1 b	56.4%
BST-122 WC	76.0 ± 14.8 b	70.5%

^1^—mean ± standard error. Different letters were used to denote statistical significance between values. ANOVA (F_3,78_ = 26.28; *p* = 7.54 × 10^−12^) and the post-hoc Tukey (*p*-value < 0.05).

**Table 6 toxins-14-00768-t006:** Effect of the BST-122 Bt strain on biomass production of different plant pathogenic fungi species.

Fungal Pathogen	Incubation Time (Days)	Biomass Surface		% Growth Inhibition	Significative Differences
		Control	Bt Treated		
*Verticillium dahliae*	6	5.17 ± 0.26	3.89 ± 0.24	24.76%	**
	12	10.46 ± 0.70	3.92 ± 0.42	63.90%	***
*Fusarium oxysporum lycopersici*	4	17.69 ± 0.89	14.96 ± 0.50	15.43%	*
	5	24.77 ± 1.07	18.48 ± 0.51	25.39%	***
	6	35.82 ± 2.43	24.08 ± 1.18	32.77%	**
	7	39.2 ± 1.96	25.98 ± 0.97	33.72%	***
*Fusarium oxysporum melonis*	4	19.32 ± 0.51	15.88 ± 0.29	17.80%	***
	5	27.30 ± 0.78	20.61 ± 0.46	24.50%	***
	6	37.34 ± 2.15	25.89 ± 1.34	30.66%	***
	7	39.70 ± 1.92	27.66 ± 1.47	30.33%	***

* *p*-value > 0.01–0.05; ** *p*-value > 0.001–0.01; *** *p*-value < 0.001.

## Data Availability

Not applicable.

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
