# Peer review of "Multifunctional Properties of a *Bacillus thuringiensis* Strain (BST-122): Beyond the Parasporal Crystal"

_toxins, 2022, doi:10.3390/toxins14110768_

Round 1
Reviewer 1 Report
This submitted manuscript deals with finding and characterizing a wild-type Bt strain to confer protection and control against diverse pests and pathogens that produce significant economic losses in agriculture. This research is quite interesting, thorough and would be a nice contribution to the literature and possibly commercial interests. The manuscript is well written. Although this reviewer has provided a significant number of specific comments to this manuscript, there is one overall major comment that most likely will require a substantial rewrite of some of the sections of this manuscript: Finding Bt organisms with multiple toxins with activity (Spore/Crystal or other compounds) against a number of various pests is not new and there are numerous book chapters written in the 1980’s that describe these: Please include statements throughout the text and in references, specific chapters in:
1) Microbial Control of pests and plant diseases 1970-1980: H.D. Burges editor. Academic Press. 1981
2) Microbial and viral pesticides. Edouard Kurstak editor. Marcel Dekker. 1982
Specific comments:
Lines 16-18: Please revise based on above comments from this reviewer
Lines 38-40: The authors are correct IF fermentation broth is centrifuged. But there are other ways of concentrating down fermentation broth (such as heat spray drying) which would retain many of these water soluble components. That is why there are Bt microbial pesticides that have quite different insecticidal activity although they contain the same AI of the same strain. Please revise.
Line 42: Generally speaking, the authors are correct, but there are clear cases of a Cry protein having toxicity against various pests: Cry1B with 3 orders of insects, Mpp51 with at least two orders of insects. Please revise
Line 46: Please include the recently published work on Mpp51 with activity against hemipterans
Line 70 and throughout the manuscript: The use of the term “biocontrol” is inaccurate as the authors are not using (or have not documented) that any of these products are actually living (from the spore standpoint, no documentation that the spore either germinates or vegetatively is growing). That is why for the most part, use of Bt is not considered biocontrol but as a microbial pesticide. Please revise.
Line 75 and throughout manuscript: Bioassaying compounds against pests are not “in vitro”. Please revise.
Lines 78-82 and in Discussion: Although it is true that there is no commercial Bt strain that controls all of the various pests as described by the authors, it is not from a lack of trying. The authors are not addressing all of the difficulties involved with commercializing such a strain, and the various media required to produce these different compounds. Note that although Bt can be cultured quite easily in fermenters, it is another thing to get Bt to grow in large populations in nature (as a true biocontrol agent). Another topic these authors need to address is that because they demonstrate multi-order and multi-phyla activity of BST-122, there most likely would need to be considerable biosafety testing on a number of organisms to assess the total spectrum of activity from an environmental safety standpoint. In other words, it appears from the introduction, that the authors wanted to find activity against their target pests, but what organisms could also be susceptible to any of the compounds in this strain? Please revise.
Lines 86-99: There is no mention of these results in Materials and Methods. Please include, including more descriptions of BST-122. Actually, the authors might want to consider writing this MS as just characterizing BST-122: What it is, what it produces, and what it has activity against.?
Lines 135-144: The authors need to state that even though Mpp51 appears to be present, the mortality observed was with Spores and crystals, therefore it is unknown exactly what killed CPB
Line 179: The authors need to note here and in Discussion that although there was 53% mortality at 100 µg/ml, this is a relatively high concentration and may not be commercially applicable.
Line 235: In planta is not the appropriate term here as the authors are conducting a root/soil drench. Please revise
Lines 321-326: There only is a two-fold difference and based on the upper fiducial limit, this reviewer is pretty sure there would be overlap of 95% CI and therefore no difference. Please delete
Line 363. Yes, 100 µg/ml concentration is less than 5 mg/ml, but still a pretty high concentration and resulting only in 53% mortality. Please revise
Author Response
Specific comments:
Lines 16-18: Please revise based on above comments from this reviewer
Lines 38-40: The authors are correct IF fermentation broth is centrifuged. But there are other ways of concentrating down fermentation broth (such as heat spray drying) which would retain many of these water soluble components. That is why there are Bt microbial pesticides that have quite different insecticidal activity although they contain the same AI of the same strain. Please revise.
We thank the reviewer for these comments related to the potentiality of Bt strains as control agents as well as secreted factors and soluble components. We have included a reference between lines 40-50 and a line in the abstract to reflect these ideas.
Line 42: Generally speaking, the authors are correct, but there are clear cases of a Cry protein having toxicity against various pests: Cry1B with 3 orders of insects, Mpp51 with at least two orders of insects. Please revise
We thank the reviewer for noticing this. We have included examples of proteins known for having a broader host spectrum between lines 55-57.
Line 46: Please include the recently published work on Mpp51 with activity against hemipterans
We have followed the reviewer’s indication and included references citing Mpp51 between lines 55-57 in the introduction section and lines 302-303 of the discussion, highlighting its activity against hemipterans and coleopterans.
Line 70 and throughout the manuscript: The use of the term “biocontrol” is inaccurate as the authors are not using (or have not documented) that any of these products are actually living (from the spore standpoint, no documentation that the spore either germinates or vegetatively is growing). That is why for the most part, use of Bt is not considered biocontrol but as a microbial pesticide. Please revise.
We thank the reviewer for the suggestion and have replaced the term biocontrol by “microbial pesticide” or “microbial control agent” throughout the text.
Line 75 and throughout manuscript: Bioassaying compounds against pests are not “in vitro”. Please revise.
We thank the reviewer for highlighting this mistake. All “in vitro” terms referring to bioassays have been corrected to now reflect “in vivo”.
Lines 78-82 and in Discussion: Although it is true that there is no commercial Bt strain that controls all of the various pests as described by the authors, it is not from a lack of trying. The authors are not addressing all of the difficulties involved with commercializing such a strain, and the various media required to produce these different compounds. Note that although Bt can be cultured quite easily in fermenters, it is another thing to get Bt to grow in large populations in nature (as a true biocontrol agent). Another topic these authors need to address is that because they demonstrate multi-order and multi-phyla activity of BST-122, there most likely would need to be considerable biosafety testing on a number of organisms to assess the total spectrum of activity from an environmental safety standpoint. In other words, it appears from the introduction, that the authors wanted to find activity against their target pests, but what organisms could also be susceptible to any of the compounds in this strain? Please revise.
We thank the reviewer for providing insight in this matter. We fundamentally agree with the above statement. Indeed this study is aimed at understanding the control properties of the toxins generated by strain BST-122 due to its gene content. Additionally, a study to address toxicity against beneficial and auxiliary insects would be required prior to any kind of product development approach. We also agree that further studies in different growth media would be necessary to improve activity against nematodes.
To reflect these comments we have made the following modifications:
Lines 87-90: “Although these preliminary results under controlled conditions could represent a good starting point in understanding the potential of strain BST-122, further studies would be required to address the precise factors, mechanisms of action and overall biosafety of the microorganism.”
Lines 455-460: “…However, further studies on BST‑122 should be performed to pinpoint the toxicological factors responsible for controlling each of the presented hosts, evaluate their activity against a broader host spectrum (for instance, hemipteran pests) and beneficial insects such as pollinators or natural enemies, soil ecosystem compatibility and address their fit at the agriculture and industrial levels..”.
Lines 86-99: There is no mention of these results in Materials and Methods. Please include, including more descriptions of BST-122. Actually, the authors might want to consider writing this MS as just characterizing BST-122: What it is, what it produces, and what it has activity against.?
We thank the reviewer for this appreciation and agree that the manuscript would improve with an additional section in the Materials and Methods (Section 5.1.1 Bacterial isolation) that explains how the strain was isolated. Additionally, we have made adjustments to text between lines in the Results section have been made accordingly between lines 113-116).
Lines 135-144: The authors need to state that even though Mpp51 appears to be present, the mortality observed was with Spores and crystals, therefore it is unknown exactly what killed CPB
We agree with the reviewer and have made changes in the text accordingly. Specifically:
Lines 153-155: “Mpp51Aa was previously reported as active against coleopterans among other insect orders [25,45], therefore, we decided to test the biological activity of the BST-122 wild-type strain against a well-known representative of this order, namely the Colorado Potato Beetle (L. decemlineata). For this purpose, we carried out bioassays in which we addressed the mortality of first-instar larvae that fed from superficially contaminated potato leaves at different concentrations of the BST-122 spores and crystals mixture”
Lines 324-328: “Although we did not determine the activity of the new Mpp51Aa protein, Mpp51Aa1 is known for being active against Colorado potato beetle (LC50=19.5 mg/ml [22,46]. Based on this previous information, we evaluated the activity of a mixture of spores and crystals from BST-122 (LC50 of 10.5 mg/ml) (Table 2). Since Mpp51Aa seems to be present in the 122GR crystals , one would assume that this protein could contribute to its overall activity against L. decemlineata (Figure 2). However, other proteins present in the crystal such as Cry5-like, Orf2_cry65A or Mpp2-like could represent an additional source of toxicity or interact synergistically with the Mpp51Aa protein.”
Line 179: The authors need to note here and in Discussion that although there was 53% mortality at 100 µg/ml, this is a relatively high concentration and may not be commercially applicable.
The reviewer is right. The concentration used in this study is too high for strain BST-122 to become an economically viable product against T. urticae. However, the aim of this study was not the development of a product based on the BST-122 strain but to understand the wide potentiality of its toxicity factors. In agreement with this, we have made modifications in lines 366-373 of the Discussion:
“However, the tested concentration was still too high from a cost-effective standpoint. Possibly, the most interesting contribution of these results is the finding of a new protein with acaricidal properties that could serve as a query for searching new Cry5-like proteins with improved activity. Additionally, a foliar application would not be effective for the control of phytophagous mites due to their feeding behavior. Therefore, opting for the expression of Cry5-like candidates in the vascular system of plants may represent a more realistic approach for the control of these pests in agriculture.”
Line 235: In planta is not the appropriate term here as the authors are conducting a root/soil drench. Please revise
We thank the reviewer for this indication. The term “in planta” has been replaced by “Activity of the BST-122 strain against M. incognita in cucumber plants” and other synonyms.
Lines 321-326: There only is a two-fold difference and based on the upper fiducial limit, this reviewer is pretty sure there would be overlap of 95% CI and therefore no difference. Please delete
We agree with the reviewer and have eliminated it .
Line 363. Yes, 100 µg/ml concentration is less than 5 mg/ml, but still a pretty high concentration and resulting only in 53% mortality. Please revise
We thank the reviewer for the feedback in this particular regard. We have included comments in the Discussion section that address this matter (lines 366-373).
Reviewer 2 Report
It is very important to find high activity strains to develop biological insecticides, and this study has certain value. And a novel Cry5-like protein proved active against the two-spotted spider mite in this report. But there's still a lot to improve. It is suggested to focus on sorting out Cry5-like protein related data and remove other unreasonable research results.
First of all, the manuscript lacks some necessary data information. This paper mainly introduces the results of strain BST-122. Including genome sequencing and insecticidal factor gene analysis based on genomic data. The article does not provide the necessary genome Data information, and the relevant genome data should be submitted to an accessible database.
In addition, the table of insecticidal factors detected in the genome only provides statistical data such as comparison consistency, which should actually provide more information. For example, the accession number, the number of amino acids in the strain protein and the number of amino acids matched in the database.
The manuscript needs to be concentrated because it contains too much meaningless experimental data. For example, in nematode assays, it is not necessary to test the mixture of bud crystals because it is difficult for nematodes to ingest larger particles. There is a similar situation when measuring the insecticidal activity of mites. The size of the ingested substance needs to be considered when feeding with the piercing and sucking mouthparts. It is unreasonable to test with the mixture of spore/crystals when the relevant situation is not clear.
The paper described that this strain was used as a biocontrol agent, and existing biocontrol products should be used as a control to determine whether the activity of this strain was comparable. There are too many strains isolated, but too few that can be used as product development applications. In addition, when the activity of mites is measured, Bt products are mainly sprayed on the surface of plants, and mite mouthparts cannot feed on Bt products on the surface, and the relevant spore/crystals mixture cannot be described as mite biological control agents. From the data, most of the assay results showed that the insecticidal activity of this strain was not high, far from meeting the requirements of product development.
There are also many problems in the determination the inhibition of plant pathogenic fungi. Compared with other species of Bacillus, such as Bacillus amyloliticus, this strain shows negligible activity. It is not appropriate to use biological control agents to describe this strain.
Author Response
First of all, the manuscript lacks some necessary data information. This paper mainly introduces the results of strain BST-122. Including genome sequencing and insecticidal factor gene analysis based on genomic data. The article does not provide the necessary genome Data information, and the relevant genome data should be submitted to an accessible database.
We thank the reviewer for pointing this out and would like to apologize for sending the manuscript incomplete. We submitted relevant genomic information to the GenBank genome database. This information is available in Materials and Methods, in the section “5.4.3. Nucleotide sequence accession number”.
In addition, the table of insecticidal factors detected in the genome only provides statistical data such as comparison consistency, which should actually provide more information. For example, the accession number, the number of amino acids in the strain protein and the number of amino acids matched in the database.
We are grateful to the reviewer for the suggestions made to improve the manuscript. Table 1 was corrected to now reflect the requested information: (before: Length (No. Residues), and now: Length (No. Amino acid residues). Three new columns were added: i) Amino acids overlap in pairwise global alignment, ii) Accession number of reference, and iii) Accession number.
The manuscript needs to be concentrated because it contains too much meaningless experimental data. For example, in nematode assays, it is not necessary to test the mixture of bud crystals because it is difficult for nematodes to ingest larger particles. There is a similar situation when measuring the insecticidal activity of mites. The size of the ingested substance needs to be considered when feeding with the piercing and sucking mouthparts. It is unreasonable to test with the mixture of spore/crystals when the relevant situation is not clear.
We thank the reviewer for this observation and have tried to condensate information and emphasize the most relevant results.
The paper described that this strain was used as a biocontrol agent, and existing biocontrol products should be used as a control to determine whether the activity of this strain was comparable. There are too many strains isolated, but too few that can be used as product development applications. In addition, when the activity of mites is measured, Bt products are mainly sprayed on the surface of plants, and mite mouthparts cannot feed on Bt products on the surface, and the relevant spore/crystals mixture cannot be described as mite biological control agents. From the data, most of the assay results showed that the insecticidal activity of this strain was not high, far from meeting the requirements of product development.
We thank the reviewer for the critical thinking regarding the possible use of this strain as a commercial product in the future. We agree that the activity levels against each of the tested target organisms is low from a cost-effective perspective. However, the aim of the study was to understand the plural potentiality of the strain based on preliminary results. Overall, the results represent a good starting point for future studies aimed at uncovering the factors behind the activity as well as their mechanism of action. Additionally, it may help drive other studies in the finding of Bt secreted factors against pests that are currently difficult to control with biological solutions, such as nematodes. We have made modifications in the text that we hope help reflect this: Lines 635-662: “. However, further studies on BST 122 should be performed to pinpoint the toxicological factors responsible for controlling each of the presented hosts, evaluate their activity against a broader host spectrum (for instance, hemipteran pests) and beneficial insects such as pollinators or natural enemies, soil ecosystem compatibility and address their fit at the agriculture and industrial levels. Overall, this work contributes to highlighting the importance of the multitarget potential of Bt strains. A notion that goes beyond the spores and crystals, by taking into consideration additional factors that could extend the use of Bt products to pests and diseases that are conventionally being dealt with by synthetic products..”
Additionally, regarding the application for the control of mites, we agree with the reviewer that foliar sprays as well as the low activity would not help controlling T. urticae populations in the field. We have made the following modifications in Lines: 366-373.
“However, the tested concentration was still too high from a cost-effective standpoint. Possibly, the most interesting contribution of these results is the finding of a new protein with acaricidal properties that could serve as a query for searching new Cry5-like proteins with improved activity. Additionally, a foliar application would not be effective for the control of phytophagous mites due to their feeding behaviour. Therefore, opting for the expression of Cry5-like candidates in the vascular system of plants may represent a more realistic approach for the control of these pests in agriculture.”
There are also many problems in the determination the inhibition of plant pathogenic fungi. Compared with other species of Bacillus, such as Bacillus amyloliticus, this strain shows negligible activity. It is not appropriate to use biological control agents to describe this strain.
We are thankful for the reviewer’s insight on this matter. It is true that in the case of Fusarium, the effect in inhibition is mild. However, for V. dahlia, the growth inhibition is around 60%. Compared with other species of Bacillus, it is true that the activity of BST-122 was lower, however, this study was aimed at exploring the potentiality of a Bt strain rather than looking for a Bacillus strain for the specific control of fungal species. Further studies would be required to have a more complete picture on the potential fungicide activity. For instance, evaluating the affected tissue in infected plants and the protection conferred by Bt-treatments.
We agree with the reviewer that the term biocontrol agent may be premature to describe the effect of BST-122 vs fungi in vitro. Therefore, we have used the term “growth deceleration” and synonyms thereof that may better describe our observations. Additionally, lines 436-440 in the discussion now reflect:
“These results suggest that the presence of the bacterium could decelerate the growth of the tested fungi species. Nonetheless, further experiments would be required to address the overall potential fungicidal activity of BST-122. For instance by infecting plant tissue to evaluate the damage caused by the phytopathogenic fungi when treated with BST-122.”
Reviewer 3 Report
Manuscript:
„Multifunctional properties of Bacillus thuringiensis: beyond the parasporal crystal“
This scientific research is a very innovative, multidisciplinary, and covers many tasks associated with modern agriculture, molecular biology, sustainable environment, gene engineering, protein biosynthesis, industrial biotechnology and bioeconomy. Diseases and pests are the major limiting factors in the production of high quality agricultural products. Bacillus thuringiensis is one of the most reliable bio-alternative to counter the agricultural losses. It is valuable to mention, that authors analyzed and found a novel Cry5-like protein, which was active against the two-spotted spider mite. In vitro and in planta assays revealed significant control of the parasitic nematode Meloidogyne incognita in the manuscript. In this work, the authors selected strain BST-122 based on its gene content and associated bioactive potential as an insecticidal, nematocidal, fungicidal, and acaricidal agent.
The manuscript is written qualitatively, the authors performed valuable experiments and this research can have many applications in the future research.
Questions and recommendations for the authors of the manuscript:
1) Please comment if you are going to apply your research results in the agriculture practice?
2) Please comment if you are you going to commercialize your products in the future?
3) Please check the whole manuscript alignment and design layout.
4) Please write about the lanes 1 and 2 in the Figure 1B (according to SDS-PAGE experiment).
5) Please correct writing M. incognita in italic style in the title of the Figure 5.
6) I can suggest writing abbreviation Bt in italic style in the whole manuscript.
7) Please check writing references according to the journal requirements.

Author Response
Questions and recommendations for the authors of the manuscript:
1) Please comment if you are going to apply your research results in the agriculture practice?
In this study, we have obtained preliminary results under controlled conditions. However, a second level to study would imply identifying the precise factors of strain BST-122 behind its activity against the various pests tested. This would provide further insights as to how developing the toxicity factors into commercial products and how their testing in greenhouse conditions would occur. Another question to consider in parallel is related to biosecurity, we need further studies to confirm that the studied strain does not affect beneficial organisms, such as pollinators and natural enemies among others. Related to the reviewer’s question, we have improved the introduction and discussion sections with some comments answered above (lines 87-90, and 456-461).
2) Please comment if you are you going to commercialize your products in the future?
With the preliminary results, we have observed that in some of the studied applications, the concentration required for treating is economically unrealistic. However, in the case of coleopteran control, this strain could be an alternative to products that can be found in the market, mainly since its gene content is totally different from the commercialized strains, hence providing variability. However, for other pests, further studies would be required to address and overexpress the pesticidal factors produced by the strain.
3) Please check the whole manuscript alignment and design layout.
We thank the reviewer for the observation and have made changes accordingly..
4) Please write about the lanes 1 and 2 in the Figure 1B (according to SDS-PAGE experiment).
We are grateful for the reviewers comment, “Lane 1: protein marker; Lane 2: BST-122 spores and crystals mixture” are now reflected in Figure 2.
5) Please correct writing M. incognita in italic style in the title of the Figure 5.
We thank the reviewer and amendments have been made.
6) I can suggest writing abbreviation Bt in italic style in the whole manuscript.
We thank the reviewer for the suggestion and have thoroughly considered it since it varies in the literature. We decided to to use Bt following a recent reference:
Crickmore, N.; Berry, C.; Panneerselvam, S.; Mishra, R.; Connor, T.R.; Bonning, B.C. A Structure-Based Nomenclature for Bacillus Thuringiensis and Other Bacteria-Derived Pesticidal Proteins. J. Invertebr. Pathol. 2021, 186, 107438, doi:10.1016/j.jip.2020.107438.
7) Please check writing references according to the journal requirements.
We thank the reviewer for observation and have inserted the references appropriately with the reference manager in the style of MDPI.
Round 2
Reviewer 1 Report
1. This resubmitted manuscript describes the BST-122 strain of Bacillus thuringiensis with activity against multiple pests in various phyla. The authors have adequately addressed most of this reviewers comments and believes that the manuscript has been greatly improved. One additional comment for consideration: Based on the revised MS and comments from the author; this MS is only dealing with the characterization of one Bt strain, and there are numerous examples of other Bt strains with various multi-pesticidal properties, this reviewer recommends that the title be revised to “Multifunctional pesticidal properties of Bacillus thuringiensis strain BST-122” or perhaps a little more provocative: “Multifunctional pesticidal properties of Bacillus thuringiensis strain BST-122: It is not just about the parasporal body”
Author Response
We thank the reviewer for the suggestions, we have changed the title and revised the text as requested.
Reviewer 2 Report
The title of the paper looks like a review, Revisions are suggested .
Author Response
We are grateful to the reviewer for the suggestions and have changed the title accordingly.